# The Effect of Utilizing Distributed Intelligent Lighting System for Energy Consumption in the Office

**Mohammed Hajjaj** *, **Mitsunori Miki and Katsunori Shimohara**

Department of Information and Computer Science, Doshisha University; Kyoto 610-0321, Japan;
mmiki@mail.doshisha.ac.jp (M.M.); kshimoha@mail.doshisha.ac.jp (K.S.)
* Correspondence: mhajjaj@mikilab.doshisha.ac.jp

**Abstract:** We have been examining how an intelligent lighting system affects energy consumption in the office. In this paper, we evaluate the traditional intelligent lighting system at the office and how to improve the best use of the intelligent lighting system by each user. The user in the large office is not able to change the level of luminance. Moreover, the traditional lighting system is not able to specify which desk is occupied or not occupied by each worker. The proper use of lighting control affects the power consumption, and the worker sometimes forgets to change the occupancy status after leaving the office, which affects the energy. This paper argues to find the best model of the intelligent lighting system to save energy by using the technology of sensing devices for detecting the occupancy of the desk. As a result, we found that each worker has the individual lighting system, and the energy of the office is reduced by improving the intelligent lighting system.

**Keywords:** distributed light system; energy consumption; automation; office

## 1. Introduction

Energy is one of the main sources that affect the intellectual productivity and performance of the employees [1,2]. In a workday, energy consumption causes concern for the designs of the workplaces and the office environment [3]. In the comfortable office, we use the intelligent lighting system to provide individual luminance for each user in the office [4]. Accordingly, we consider the problem of high energy consumption for the intelligent lighting system used in the office.

A lot of technological advancements have been applied in the workplace to enhance how the lighting control system performs [1,5]. The advanced model of lighting control encompasses how the system provides comfortable dynamic light for each user besides improving the high energy consumption of the office [6]. The available technological solutions in commercial environments do not adequately capture the relationship between energy efficiency and visual comfort of the luminance [7]. For instance, the typical improvements include how to decrease the energy consumption of the intelligent lighting system using remote sensing technology and utilizing the automation process on the system [8].

One of the advance models for the lighting control system is to involve sensing technology within the lighting system [1]. The sensing device technology is used in general as an interface between the user and the lighting control system. In the intelligent lighting system, the user can communicate with the lighting control using the illuminance sensing device [9,10]. Additionally, the automation process is another advanced technology utilized for the intelligent lighting system to make the automation method more practical and dynamic based on the user requirement [1,9].

The sensing device technology and the automation process enable the intelligent lighting system to become more useful and dynamic for each user in the office [11]. For example, each worker feels free

to set the illuminance target in the desk using the sensing device, while the system obtained the target using the automation dimming process of the luminance. In this case, the system influences the energy consumption of the lights in the office.

In the lighting control system, the illuminance comes from luminance, which is the luminous intensity reflected off the surface of the desk [10]. The illuminance is the amount of reflected light on the desk [10]. Each luminance has the amount of energy based on the value received by the sensing device on the desk [12]. The concern of this traditional method is how sensing devices are used to influence the energy consumption of the office. The proper use of sensing devices affects how the intelligent lighting system performs in the office [4]. The sensing devices have two functions for the system [10]: at first, toggle the status of the lighting system and initialize the system to connect with the sensing device, then the second functionality is to change the target illuminance of each user in the desk. The user determines the occupancy status of the desk by toggling the status of the sensing devices [13]. Each user changes the luminance as desired while the sensing device sends the single to the intelligent lighting system, and the system provides the individual illuminance based on the sensing devices [3].

The intelligent lighting system detects the area at the office by checking the occupancy status of each worker [3,13]. Therefore, the system received the value of the sensing devices while the dimming process of the lighting system launched to provide the target illuminance for each worker [13].

In this paper, we have been examining how the intelligent lighting system performs using wireless technology. For instance, we are going to implement the intelligent lighting system with Bluetooth Low Energy or BLE beacons to toggle the occupancy status of the desk [3,13]. The beacon device is a model of the wireless technology standard to broadcasting a short message to other devices [6].

The significance of this paper may lie in the use of a simple sensing device beacon connecting with a smartphone to toggle the occupancy status for each worker. The beacon is easy to install and relocate at the office, and the sensing device is not expensive in comparison with other methods of developing an intelligent lighting system. In this way, the intelligent lighting system provides the luminance based on the current status of the user at the office. Also, other unoccupied desks are not required to be active so that the intelligent lighting system reduces the energy of the environment.

The rest of this paper is organized as follows. The next section highlights some of the related works. After that, this paper describes the models of the system and the proposed solution of the intelligent lighting system. Then finally, the results of the experiments discussed the performance of the intelligent lighting system.

## 2. Related Works

Direct illumination can often be expensive to compute, particularly when multiple lights are involved. There have been a variety of techniques proposed to reduce the cost of energy. In this section, we briefly discussed the existing models of the intelligent lighting system in the case to observe the rate of energy consumption in the office. However, there are a lot of works used to track the positions of users inside the workplace.

The authors of [4] used the wireless sensing networks to implement some solutions for reducing unnecessary energy consumption of the smart houses. The paper has a method to analyze the illumination values to evaluate the energy consumption of the lighting systems, and they found that the energy reduced with the lighting control system.

The authors of [5] argued the strategies that reduce the needs of the intelligent lighting systems, and the study found that a change in occupancy status of the workers leads to an increase in lighting energy use. However, the work in [1] determines the significance of using the automation control process of luminance to save energy for each worker in the office.

The paper [8] has a real-time locating system, and the authors use the system based only on the BLE technology to support interactive communications of beacon devices and smartphones. The authors combine the broadcast and mesh topology options to extend the applicability of beacon solutions to detect the presence of specific users at specific locations, and then the present state can be sent to the application server via the relay of beacon devices.

Additional studies were conducted to implement beacons functions, such as information communication and sensors in the lighting control system [13]. The authors have some models using beacon technology to observe the distance proximity to examine the influence on the power consumption of the system in the office. The authors of [2] highlighted the problem of energy consumption by the lighting control system and proposed design practices to avoiding energy waste by the lighting system.

The authors of [3,8] proposed an intelligent lighting system using a BLE beacon to toggle the occupancy status of the workers. The paper discussed how the new model of the intelligent lighting system performs to realize the target for each user in the workplace.

We have extended our work [10] and developed a conventional method of lighting control using new technology for tracking the worker at the office. This study argues the impact of using an intelligent lighting system in a real office and how the system becomes more dynamic and efficient for the office.

Generally, some works present different aspects of the office environments about examining the occupancy status of the desk. Also, the papers work on the luminous conditions and location tracking of the user at the workplace. The purpose of the works is to verify the impact of energy to enhance the performance of the office.

## 3. The Architecture of Lighting Control Model

The intelligent lighting system consists of the basic components of a ceiling lighting fixture, in addition to the controller unit, and an illuminance sensing device. We have replaced the illuminance sensing device with another technological system rather than the manual control in the system. In this section, we describe the conventional model of the intelligent lighting system, and the new model of the system, as shown in Figure 1.

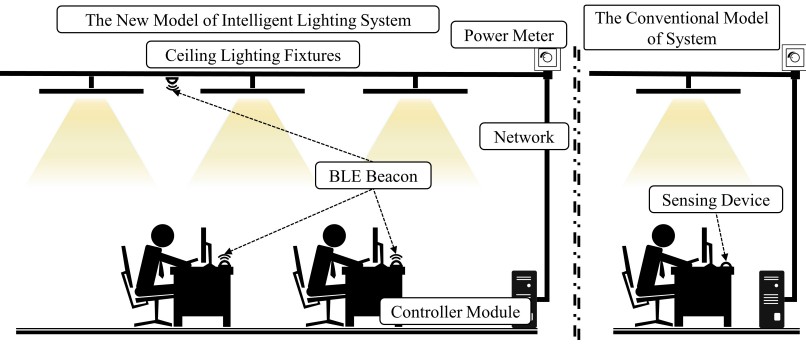

**Figure 1.** The intelligent lighting system in two designs. The traditional system on the right side, and on the left side, the new design of the system.

### 3.1. The Conventional Intelligent Lighting System

In the conventional model of the lighting control, the system consists basically of a dimming unit of luminance called the controller [10]. The controller is connected by each illuminance sensing device, which has been used by each worker to change the illuminance target on the desk. After that, the controller sends the process of the dimming light to the power meter after receiving the values from the user to make sufficient luminary based on the user requirement.

The intelligent lighting system is a part of the automation lighting system. Consequently, the optimization formula is used to achieve the target for each user derived basically from the illuminance and power consumption as in Equation (1).

$$f_i = w_p \times p + w_g \times \sum_{i=1}^{n} g_{ij}$$
(1)

$$g_{ij} = R_{ij} \times (Ic_j - It_j)^2$$
(2)

$$I = \sum_{i=1}^{n} R_{ij} L_i$$
(3)

where $f$: objective function of light, $p$: power consumption, $g_{ij}$: illuminance constraint, $w$: weighting factor, $i$: light, $j$: sensor, $R_{ij}$: influence coefficient, $Ic_j$, $It_j$: current and target illuminance. $L_i$: luminous intensity of candela in the desk.

The intelligent light system changes the illuminance pattern for each desk individually based on the target. In a short time, the system obtains the target for each worker using the automation method at the office. We replace the sensing device for each desk, and we propose a new solution for the system using BLE beacon. The beacon will install instead of using the illuminance sensing device on the desk. Another beacon will install in the ceiling lighting fixture to improve the accuracy and enhance the performance of the system. The beacon attached to the smartphone of each worker is the interface that can connect to the system. In this case, each worker has the interface to set or adjust the desire of the illuminance target.

### 3.2. The System Setting

The system composes from the sensing device of the smartphone with beacon technology to toggle the occupancy status of each user. BLE beacons transmit the signal radio waves, and the smartphone receives the signal from the beacons [8]. The data transferred includes an alerting signal mode of the device [3,4].

A BLE beacon is a model of a one-way wireless technology standard to broadcasting a short message to other devices [6]. The message includes information about the device identification number and the services offered to other configured devices nearby [3]. We are going to use this solution embedded in the intelligent lighting system to improve the occupancy status and detect the certain desk of worker proximity [3].

The signal strength transmitted by the beacon has determined the occupancy status of the desk in the office. For instance, each user has one beacon in the desk and another active beacon placed in the ceiling fixture of the office for efficiency measurement. Consequently, the intelligent lighting system has the accuracy of determining the occupancy status of each user at the office. The lighting system lies to provide the proper individual luminance for each user based on the occupancy status for each desk, which affects the energy consumption of the office.

The beacon used radio-frequency waves to transmit the signal values, including an information packet about the device in a short time and the regular interval range. Then, the smartphone estimates the distance from the beacons by using the information packets and received signal strength [3].

The received signal strength measured the signal energy for each received packet to quantize the received signal strength indicator (RSSI). The proposed intelligent lighting system works on the RSSI of the smartphone to determine whether the seat is occupied or unoccupied. RSSI identifies four parameters: range, accuracy, linearity, and averaging period [3].

Therefore, the RSSI range is the smallest and largest transmission signal in dBm, in which the accuracy is the average error for each transmission signal. The indicator linearity deviates the computed function of energy. RSSI is changed by the distance between the beacon and the smartphone, as in Figure 2. RSSI is measured to know the changes in the distance between the beacon and the smartphone.

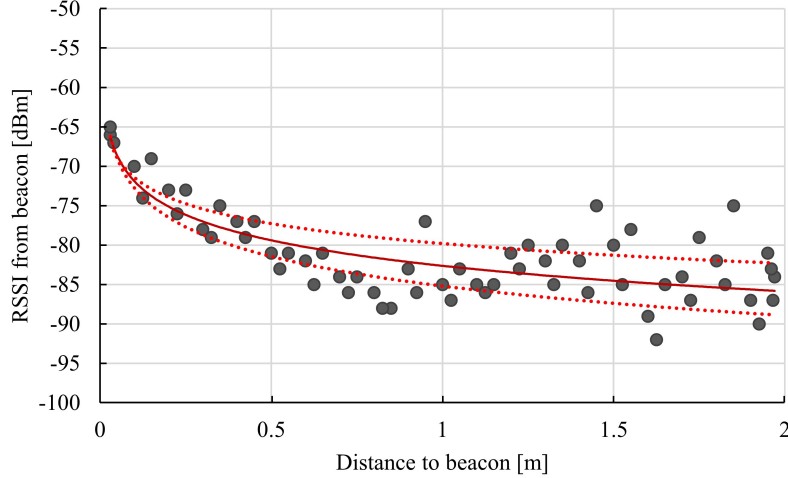

**Figure 2.** The received signal strength indicator (RSSI) experiments of beacons.

The RSSI computed in the receiver to the transmission energy indicates broadcasting by the beacons [3]. The RSSI is affected by the location of the beacon and the mobile. Then, the mobile sends a signal to the control module of the intelligent lighting system to occupy the desk for the worker [3].

## 4. The System Experiments

In this experiment, we have 42 positions for the workers, as shown in Figure 3. Each desk has a random target of 300 lx, 500 lx, and 700 lx for the preferred illuminance. We distribute the illuminance based on the preferred individual illuminance for each user in the office.

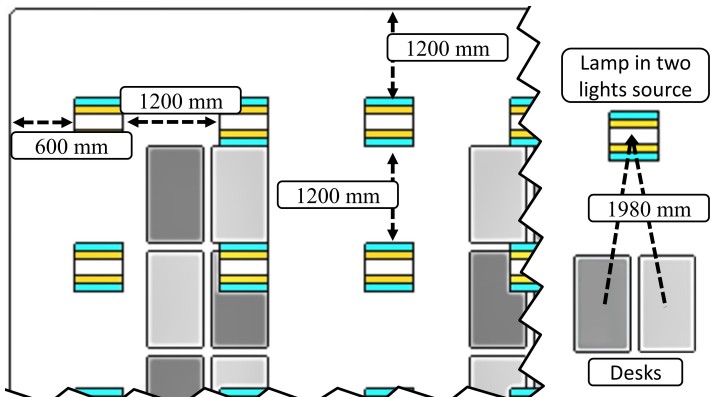

**Figure 3.** This is a part of the experiment, including 42 lights and 42 desks.

In the actual office design and traditional lighting control, the worker cannot change the illuminance. All the workers have the same level of standard illuminance as recommended. In the conventional intelligent lighting system, each worker can choose the level of the individual illuminance on the desk.

The preferred illuminance was taken by a survey conducted in an actual office in Tokyo. In the traditional lighting control, each worker has 700 lx as a standard level of illuminance, while in the conventional intelligent lighting system, we found that around 20% of workers have the illuminance at the level of 300 lx. Furthermore, we found that another 20% of workers have the illuminance of 700 lx each, while around 60% of workers get the level of 500 lx.

In our experiments, we observed the energy effect and monitored the impact of using intelligent lighting systems on the performance rate of power consumption in the office. In the conventional intelligent lighting system, toggling the sensing devices have influenced the use of the lighting system.

For instance, we assume the system has two scenarios for the status of the system affected by the usage of sensing devices during the daytime.

The first scenario of our experiment is related to the toggling status of the desk and sensing devices. The ratio of the occupancy status of the desk is the number of workers that used the intelligent lighting system and never left the office in the daytime. In this scenario, the intelligent lighting system has several cases when the workers are not in the office. We assume 30%, 60%, and 90%, or all of the workers leave the office, respectively. We have this assumption to study the behavior of the intelligent lighting system in the normal case and to monitor the effect of power consumption in different cases. For each case of the toggling status of the system, the purpose of the assumption will lead to check the overload of the energy in the office, regardless of the power consumption in the normal case of using the intelligent lighting system in the office.

The second scenario of the experiments checks the overload of the energy rate in the intelligent lighting system through the proper use of the sensing devices. The sensing devices toggle the status of the system by each worker before receiving the values. Toggle the status of the sensing device is done manually by each worker in the office. The proper use of sensing devices keeps the lighting system active for each worker while the worker was using the system before leaving the office. Whatever the intelligent lighting system has an active status or a nonactive status, and the status depends on the toggling of the sensing device.

Occasionally, each worker activates the intelligent lighting system during work time or deactivates the system due to forgetting to toggle the status of the system in the office. Consequently, the ratio of forgetting to toggle the status of the intelligent lighting system is the number of workers to the total who neglect the sensing devices while leaving the office.

Regardless, we assume, in our experiments, different cases. We have 20%, 40%, 60%, and 80% of workers for the case of forgetting to toggle the system after leaving the office, respectively. We conducted the experiments to study the behavior of using the sensing device properly in different cases and the effect of energy consumption in the office. In the experiments, we had 42 workers and desks for each to use the intelligent lighting system. The energy of the intelligent lighting system becomes in full use of power in the case that the ratio of forgetting to toggle the status of sensing devices is 0%. In another case, the ratio of the occupancy status is 70%, while 30% of the workers leave the office, which assumes that 29 workers are still using the intelligent lighting system.

The overloaded energy status of the intelligent lighting system becomes higher than before the previous status based on the ratio of the forgetting status to toggle the sensing devices. For more clarification, we assume the occupancy status of total workers is 10%, which corresponds to 38 workers having left the office while the intelligent lighting system works.

However, if we assume the ratio is 60% for the forgetting status of toggle the sensing devices, it indicates that 23 workers left the office and forgot to change the deactivate the intelligent lighting system. The energy consumption, in this case, will increase, and the power overload is caused by the occupancy status of the desks. Furthermore, the system provides the energy for 27 workers, while the energy for this case is required for four workers. This case indicates that energy will become higher than before in the previous case.

We conducted our experiments by using the previous scenarios, as depicted in Table 1. The table shows the ratio of occupancy and the ratio of forgetting toggle the status of the system for all the workers in the office. For instance, if the ratio of occupancy is 100% or the ratio of forgetting to toggle the status of the sensing devices is 100%, the total number of active illuminance sensing devices is 42 as well. In this case, the energy consumption will become high for the system.

Table 1 shows, for example, if the occupancy status of the office is 10%, it assumes that there are four workers using the intelligent lighting system in the office. On another side, if the ratio of forgetting to toggle the status of the system is 40% of the workers who left the office, it assumes that around 15 workers kept the sensing device in the active case. The overload of the energy in the previous figure increased by four times the normal case.

**Table 1.** The ratio of occupancy and the ratio of forgetting to toggle the status of the system for all the workers in the office.

| Cases | | The Ratio of Forgetting to Toggle the Status of the System | | | | | |
|---|---|---|---|---|---|---|---|
| | | *0%* | *20%* | *40%* | *60%* | *80%* | *100%* |
| **The Ratio of Occupancy** | *100%* | 42 | 42 | 42 | 42 | 42 | 42 |
| | *70%* | 29 | 32 | 34 | 37 | 40 | 42 |
| | *40%* | 17 | 22 | 27 | 32 | 37 | 42 |
| | *10%* | 4 | 12 | 19 | 27 | 35 | 42 |
| | *0%* | 0 | 8 | 17 | 25 | 34 | 42 |

In the experiment, we had 42 workers, and around 20% of workers had 300 lx and another 20% of workers had 700 lx each, while 60% for the workers had 500 lx for each. The intelligent lighting system tries to realize the target for each worker, and the energy consumption becomes as in Figure 4. The energy overload reached the maximum power in the office. The energy rate becomes stable in around three to five minutes to realize the target for each user. Therefore, the energy rate becomes stable in a very short time as well when the situation of the occupancy ratio is changed from 100% to 70%, which is applied for all the situations for the occupancy status.

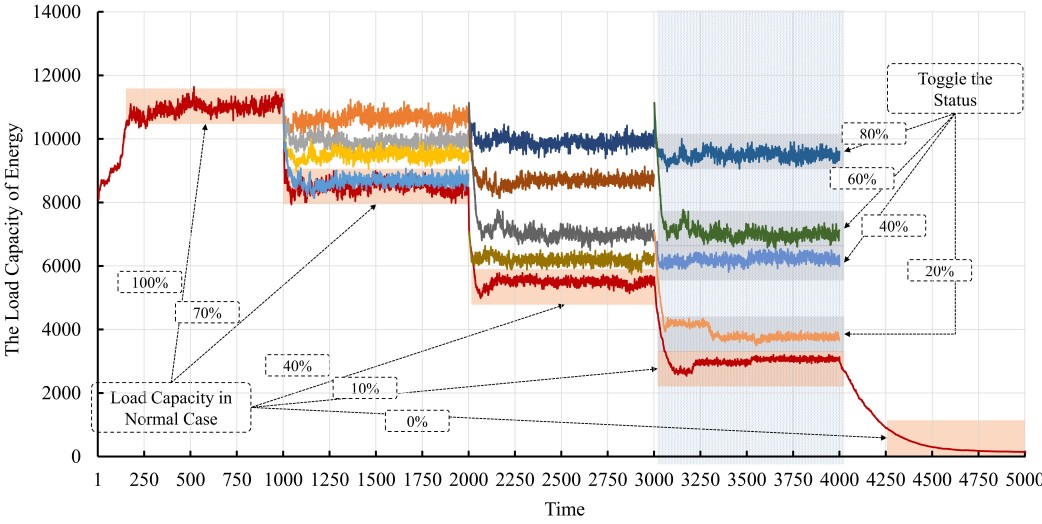

**Figure 4.** The history of the energy consumption at the office.

The ratio of forgetting to toggle the status of the system affected the energy consumption rate. The energy overloads more than the required energy for the occupancy status. To further illustrate, the occupancy status has around 3000 watts for 10% of workers inside the office, while 90% of workers deactivate the system after leaving the office. However, the ratio of energy consumption increased to 100% in the case where 10 of the workers were not using the intelligent lighting system properly. Also, the ratio will increase more to 200% if the number of workers doubles.

The occupancy status and the forgetting status of toggling the intelligent lighting system have a big influence on the energy consumption of the office, as depicted in Figure 5. The occupancy ratio and the forgetting ratio of toggling the status of the system have a relationship with affecting the energy consumption rate in the office.

Utilizing the lighting control system by beacons instead of the traditional system tries to decrease the rate of energy consumption and fix the problem of activating the illuminance sensing devices manually. The proposed solution depends on the number of workers in the office as the beacon has a limited range of detecting the signal of smartphones and how wide the office is for workers. For instance, we used two types of beacons: each user has one beacon at the desk, and another beacon installed in the ceiling lighting fixture to detect the movement of the worker inside the office.

Each beacon detects the signals of the smartphone by the distance and sends the received signal to the intelligent lighting system. Each worker will use the smartphone as an interface to change the target illuminance.

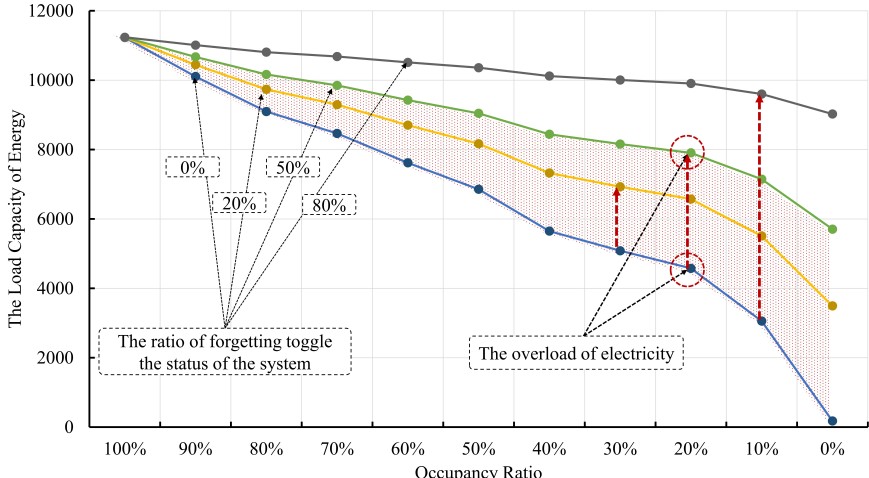

**Figure 5.** The load capacity of energy for the occupancy status compared with forgetting to toggle the status of the intelligent lighting system.

In the next experiments, we have four workers to validate the intelligent lighting system in realizing the target for each worker. The illuminance targets 350 lx, 500 lx, 400 lx, and 600 lx are distributed for each worker, respectively. Each worker changes the illuminance target using the smartphone, and the smartphone is close to the beacon on the desk. Based on RSSI, the smartphone received the signal from the close beacon. Then the system detects the beacon and the intelligent lighting system realizes the target for each worker. The target was realized as 378.76 lx, 494.24 lx, 383.41 lx, and 581.45 lx for each worker, respectively, as depicted in Figure 6.

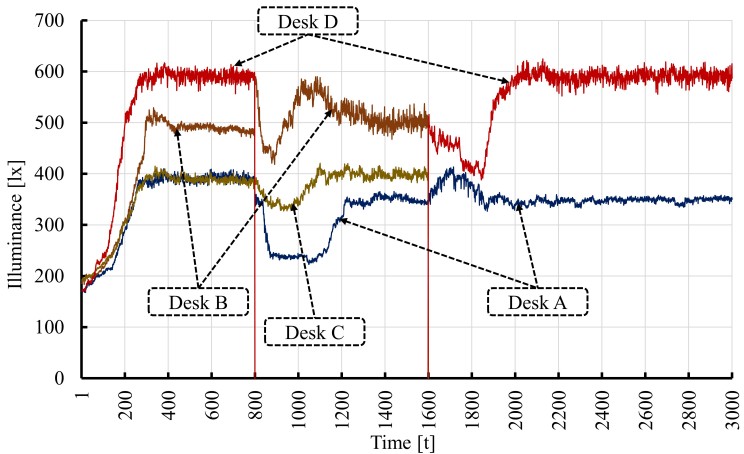

**Figure 6.** The illuminance history of the sensor devices is during the different status of the occupancy inside the office.

After that, each worker moves to change the position inside the office. For example, worker A moved to different places inside the office in a short time, and the worker came back to the desks. The workers B and C moved as well to different places in a short time, and then they left the office. Also, worker D left the office in a short time and came back later to the office.

Based on this scenario, the intelligent lighting system with beacons works properly, and the system was activated based on the user movement inside the office. In the first case of the workers,

the intelligent lighting system does not detect the signals of beacons in the desks received by the smartphones of the workers. However, the system detected the signals of beacons in the ceiling lighting fixture received by the workers A, B, and C, while the system did not receive the signal of worker D as the worker left the office.

The main purpose of the beacon on the ceiling lighting fixture is only to detect the user movement inside the office. After the workers B and C left the office, the system did not detect the signals of beacons in the ceiling lighting fixture received by the workers B and C. However, worker D came back to the office, and worker A went back to the desk, and the intelligent lighting system changed based on the user movement. The system detected the signals of beacons in the desks received by the smartphones of the workers A and D.

The intelligent lighting system behavior changed dramatically based on the users needs and movement. The system provides the target illuminance for each user in the office, while the system controls the toggle of the status of the illuminance sensing devices based on the user movement. In this solution, the system commands the energy of the lighting control system and changes the luminance based on toggling the occupancy status of the sensing device at the desk.

## 5. Conclusions

This paper introduced a new model of the intelligent lighting system using beacons for toggling occupancy status in the office. The new model works to improve the rate of energy consumption of the office using advanced technology to track the worker inside the office.

Utilizing sensing technology is conducive to improving how the system performs for each worker in the office, and the intelligent lighting system serves to provide the required illuminance in the office. However, sensing technology is widely applied in a different perspective in office design.

We found in the actual office design that the workers cannot change the standard illuminance provided on the desk, and the lighting control system is not flexible enough to change the level of illuminance. In the conventional intelligent lighting system, we found around 80% of workers have changed the standard level of illuminance and preferred different levels of the individual illuminance instead to make the desk more comfortable for the human eye.

Therefore, utilizing the illuminance sensing devices affects the intelligent lighting system and causes concern for each worker in the traditional intelligent lighting system. We found that the ratio of the forgetting status of toggling the illuminance sensing devices affected the rate of the energy consumption of how the system performs for each worker in the office.

The rate of energy consumption decreased compared with the traditional method of the intelligent lighting system and the new method of the system using wireless technology in the office. Furthermore, we found that the occupancy status affected the rate of energy consumption in the office.

The proper use of illuminance sensing devices affects energy reduction in the conventional intelligent lighting system, and in the new model, the energy is reduced more by using beacon technology. We found that the new design of the intelligent lighting system using beacons inside the office contributes to the energy-saving of the office design.

As a result, the proposed system is operated properly in the office and each worker has the preferred individual illuminance inside the office during the work time at the desk, which becomes more useful, dynamic, and effective when introduced in the real office design.

**Author Contributions:** Conceptualization, M.H., M.M., and K.S.; methodology, M.H.; validation, M.H., M.M. and K.S.; formal analysis, M.H.; investigation, M.H., and M.M.; data curation, M.H.; writing–original draft preparation, M.H.; writing–review and editing, M.H.; supervision, M.M. and K.S. All authors have read and agreed to the published version of the manuscript.

**Funding:** This research received no external funding.

**Conflicts of Interest:** The authors declare no conflict of interest.

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
