# Peer review of "The Effect of Utilizing Distributed Intelligent Lighting System for Energy Consumption in the Office"

_applsci, doi:10.3390/app10062004_

Round 1
Reviewer 1 Report
How to obtain comfortable illumination is a very important issue, especially in the workplace. When office lighting is combined with a reduction in energy consumption, it is a good example of how exploiting advanced technologies for a better life.
The concept is well developed in a research validated by suitable experimentation.
The illustration of the research work interesting and clear., with an appreciable synthesis in the explanations and useful details about the tests and results.
The conclusions should be expanded. I possible additional aspect to be discussed could be the standards and norms for lighting in workplaces, which depend on the country of application and could affect the solution for office illumination.
English is correct and the text is clear.
Author Response
Dear Reviewer,
Thanks so much for the comments.
According to your comments, I am writing to you here:
Point 1: The conclusions should be expanded.
Response 1: We consider this point and expand our conclusion in more detail a little bit.
Point 2: I possible additional aspect to be discussed could be the standards and norms for lighting in workplaces, which depend on the country of application and could affect the solution for office illumination.
Response 2: We consider this point and explain the standard level between the current system, the old one, and the proposed model. Our experiments have been done in Japan.
Thanks so much for the feedback, and I am looking forward to hearing from you again.
Much appreciated,
Reviewer 2 Report
Although the work seems to have been carried out rigorously, it is let down by the way it is presented. The use of certain terms (e.g. Modular) and the mixing between luminance and illuminance as well as the use of improper prepositions (to, for etc..) makes it difficult to make sense of.
I would recommend a thorough text editing using a professional proof reader if necessary.
Author Response
Dear Reviewer,
Thanks so much for the comments.
We consider all the points, and according to your comments, I am writing to you here:
Point 1: The use of certain terms (e.g. Modular).
Response 1: We consider this point and change it to other terms based on the paper which becomes more easier for readers.
Point 2: The mixing between luminance and illuminance.
Response 2: We explain a little bit here about the concept and the difference between illuminance and luminance to make it easy for the reader. We added the definition of the concepts here in the paper.
Point 3: The use of improper prepositions (to, for etc..) makes it difficult to make sense of.
Response 3: We consider this point. I am afraid if there is something confused here, we check it out, so please would you indicate to us some of them.
Thanks so much for the feedback, and I am looking forward to hearing from you again.
Much appreciated,